# Correlation between Endometriosis and Selected Allergic and Autoimmune Diseases and Eating Habits

**DOI:** 10.3390/medicina58081038

**Published:** 2022-08-02

**Authors:** Aleksandra Nowakowska, Katarzyna Kwas, Angelika Fornalczyk, Jacek Wilczyński, Maria Szubert

**Affiliations:** Clinic of Surgical and Oncologic Gynecology, Department of Gynecology and Obstetrics, Medical University of Lodz, M. Pirogow Teaching Hospital, Wilenska 37 St., 94-029 Lodz, Poland; katarzyna.kwas@stud.umed.lodz.pl (K.K.); angelika.fornalczyk@stud.umed.lodz.pl (A.F.); jrwil@post.pl (J.W.); maria.szubert@umed.lodz.pl (M.S.)

**Keywords:** endometriosis, allergic diseases, autoimmune diseases

## Abstract

*Background and Objectives*: Endometriosis is a hormone-dependent chronic inflammatory disease with serious reproductive and general health consequences. It is viewed as a multifactorial problem, consisting of matters related to altered immunity and genetics. In this study, we determined the correlation between endometriosis and allergic and autoimmune diseases in patients at reproductive age. *Materials and Methods*: Online surveys distributed through websites related to gynecological problems. The questionnaire was composed of 63 single and multiple-choice questions concerning the course of endometriosis, diet, and allergic and autoimmune diseases. The obtained data were assessed using statistical tests. *Results*: 501 female patients (mean age 31.1 SD = 6.8) were included in the study. The control group (*n* = 155) consisted of healthy females, whereas the study group (*n* = 346) consisted of female patients with endometriosis; each group was subdivided according to allergy status. There were statistically significant differences between groups for the following: positive family history of endometriosis (*p* = 0.0002), onset of allergic symptoms (*p* = 0.0003), frequency and duration of abdominal pain (*p* = 0.00625), and defecation disorders (*p* = 0.0006). Asthma was less common in the study group (*p* = 0.00611). The group of patients with endometriosis and allergies had a high median of consumption of red meat (*p* = 0.0143), fish (*p* = 0.0016), and dairy products (*p* = 0.0001). *Conclusions*: Endometriosis did not affect autoimmune diseases and their courses. Patients with diagnosed endometriosis presented allergy symptoms much earlier than the healthy patients. The consumption of dietary products such as soya products, red meat, and alcohol had an influence on the occurrence of endometriosis.

## 1. Introduction

Endometriosis is a hormone-dependent chronic inflammatory disease with serious reproductive and general health consequences [1]. The classical definition of endometriosis includes the presence of endometrial glands and stroma in ectopic locations, primarily the pelvic peritoneum, ovaries, rectovaginal septum, and uterosacral ligaments [2]. The main symptoms are dysmenorrhea, dyspareunia, chronic pelvic pain, irregular uterine bleeding, and/or infertility. This pathology applies to around 6–10% of women, with the highest incidence occurring for patients between 25 and 35 years old, contributing significantly to a reduction in fertility. Although the relationship between endometriosis and infertility as a definitive cause−effect connection has been clinically recognized and proven, and has been well described in the literature, it remains unclear [3]. Currently, endometriosis is viewed as a multifactorial disease, consisting of issues related to altered immunity and genetics, which affects not only the fallopian tubes and embryo transport, but also the normal endometrium [3,4]. The etiology of the disease is not fully understood, but immune, endocrine, genetic, and anatomical disorders have been mentioned as risk factors [5,6]. The factors contributing to endometriosis include inflammation (overexpression of chemokines, matrix metalloproteinases, cytokines, and prostaglandins), angiogenesis, oxidative stress, apoptosis resistance, and immunological diseases. In recent years, the theory has been developed that the origin of ectopic endometrial tissues may be connected with abnormalities in the immune system, and deficiency in cellular immunity in women with endometriosis [7]. This is the reason endometriosis is associated with other diseases, including allergic diseases [8]. There is also evidence suggesting that specific types of dietary fat are associated with a higher risk of endometriosis [9,10]. Diet and lifestyle can impact the risk of developing endometriosis, which justifies the fact that nutrition may influence the presence of inflammation in the body [11,12,13]. A lack of understanding about the pathophysiology of endometriosis and its correlation with other diseases affect the limited management. Therapy is based on GnRH analogues, combined oral contraceptive, anti-progestogens, and sometimes surgical treatment. In the future, studies may present a new path for the treatment of endometriosis. Making minor but critical changes, such as eating habits or diagnosing other correlated diseases, can significantly improve patients’ quality of life.

We conducted this study to examine the relationships between endometriosis and allergic and autoimmune disorders and eating habits in the general population of women by using the internet and social media.

## 2. Materials and Methods

In this research, we analyzed the correlation between endometriosis and selected allergic and autoimmune diseases and eating habits, using a questionnaire study. The research was conducted over the internet, using social networks and support groups. The survey was aimed at women diagnosed with endometriosis, as well as women not suffering from endometriosis but who have allergic symptoms or are completely healthy.

The survey was anonymous, and each participant was informed about this fact and about the objectives of the study. Each participant was asked to give informed consent to participate in the study by ticking an agreement on the website survey page. Then, each patient was asked to provide the date of their diagnosis of endometriosis and the method of diagnosis. A detailed description of the diagnosis was mandatory for further analysis among the endometriosis patients. Questionnaires with empty answers were excluded from further analysis in order to avoid bias resulting from misinterpretation. Exclusion criteria also included refusal to consent to the examination, as well as being younger than 18 years old..These data were obtained as a result of detailed questions that were only open if the respondents checked “yes” for endometriosis diagnosis. The principles of the Declaration of Helsinki were followed when carrying out the present study.

The questionnaire consisted of 63 single- and multiple-choice questions concerning endometriosis, allergic and autoimmune diseases and symptoms, eating habits, comorbidities, gynecological and obstetric history, and lifestyle. 

The data from the completed surveys were analyzed using Statistica Version 13.3. Data were checked using the Chi^2^ test for non-parametric variables, ANOVA Kruskal−Wallis test, and Mann−Withney U test. Continuous data that did not fit a normal distribution were described as median (interquartile range). Categorical data were described as frequency (percentage). The study was approved by the local Ethics Committee—RNN/334/19/KE.

## 3. Results

Here, 501 female patients were included into the study. The study group was composed of patients with diagnosed endometriosis (*n* = 346) with a mean age 32.48 years (SD = 5.97). Among the study group, patients with allergies and those without were present. The division of groups in this study is presented in Figure 1. 

The control group consisted of patients without endometriosis (*n* = 155) with a mean age of 27.39 (SD = 6.91), where 81 females (52.3%) reported allergies and 74 patients (47.7%) reported no allergy diagnosis. The characteristics of the groups regarding their parity status are presented in Table 1. The diagnosis of the individuals in the study group (patients with endometriosis) was stated by the patients in the survey, where 49.6% of them had a laparoscopy (36.3%) or laparotomy (15.3%) performed. The remainder of the females had transvaginal ultrasonography (27.5%), contrast-enhanced ultrasound (15%), or magnetic resonance imaging of the lower pelvis (5.2%) performed, or were diagnosed during an advanced hysteroscopy (<1%) or cesarean section (<1%). Both groups were also asked whether they used to smoke. The difference in frequency of smoking between the study and control groups was not statistically significant (*p* = 0.21177). Obstetrical complications occurred in 6.7% (*n* = 34) of all patients. The difference in several obstetrical complications between the group of patients with endometriosis with allergies and the healthy individuals with allergies was not statistically significant (*p* = 0.84358). Here, 51.2% of patients with endometriosis and only 44.4% of healthy patients without endometriosis had diagnosed allergies, but this difference was not statistically significant (*p* = 0.3018). An analysis of the influence of comorbidities on endometriosis occurrence showed that none of the following diseases had a significant affect: hyperthyroidism (*p* = 0.6632), hypothyroidism (*p* = 0.1214), Crohn disease (*p* = 0.2733), coeliac disease (*p* = 0.1208), or diabetes type 1 (*p* = 0.491). 


Smoking


Both groups were asked whether they used to smoke; 13.8% of patients confirmed smoking (17% of the study group and 15.6% of the control group). None of the patients smoked more than two packs of cigarettes daily, and only nine patients stated using e-cigarettes. The difference in frequency of smoking between the study and control groups was not statistically significant (*p* = 0.21177).


Obstetrical complications


Obstetrical complications occurred in 6.7% (*n* = 34) of all patients. The difference in several obstetrical complications between the group of patients with endometriosis with allergies and the healthy patients with allergies was not statistically significant (*p* = 0.84358). 


Allergies


An analysis of the food allergy for nickel, and as well as of cobalt allergy, was performed.

It was shown that 51.2% (*n* = 105) of the study group and 44.4% (*n* = 36) of the control group had food allergies (*p* = 0.302). Moreover, it was stated that patients with diagnosed endometriosis presented allergy symptoms much earlier than the healthy patients (*p* = 0.0003). However, allergic rhinitis was presented by 55.12% of patients with endometriosis and 62.96% of healthy patients. The difference between the patients was not statistically significant (*p* = 0.227).

The analysis of the most common allergies also showed no difference in the frequency of the nickel allergy (21% of the study group vs. 14.8% of the control group) and of cobalt allergy (11.2% of the study group vs. 14.8% of the control group). 


Food allergies


In a further analysis, variables concerned with the consumption frequency of soya products, beef, poultry, fish, and dietary products, as well as alcohol use, were considered. The frequency of soya product consumption was significantly correlated with the endometriosis diagnosis (*p* = 0.008). In addition, a significant correlation between the frequency of alcohol use and endometriosis occurrence was stated (*p* = 0.0293).

The median poultry consumption was similar in all four groups. However, healthy patients were more balanced (*p* = 0.0145). The highest median of beef consumption was present in the group of allergic patients with diagnosed endometriosis. An analysis of the consumption of fish showed that the median was similar in all of the groups. However, the study group was more balanced (*p* = 0.0016). The differences between the study and the control groups for beef consumption were statistically significant (*p* = 0.0143). 

The analysis of the consumption of dairy drink products showed that the median was the smallest in the group of allergic patients with diagnosed endometriosis (*p* = 0.000). The analysis of the white cheese showed that the median consumption was similar in all four groups. Yellow cheese consumption was the smallest in the group of allergic patients with diagnosed endometriosis, and the difference was statistically significant (*p* = 0.0002). 


Gastrointestinal tract symptoms


Patients answered questions regarding abdominal pain, perianal lesions, bowel movement disorders, mouth lesions, and body mass loss. Abdominal pain occurring earlier than 6 months ago and lasting more than one day per week was significantly more common in the study group. Furthermore, the differences in bowel movement changes, perianal lesions, and abdominal pain resolving after defecation were also statistically significant. The distribution of answers is presented in Table 2.


Asthma


Here, 11.7% of patients with endometriosis and 24.7% of healthy patients had coexisting diagnosed asthma. The difference between the study and control groups was statistically significant, meaning that asthma was more common in patients without endometriosis (*p* = 0.00611).

However, there was no difference in the frequency of symptoms (25% of patients with endometriosis vs. 30% of healthy patients; *p* = 0.7108), in limiting daily activities (41.67% of the study group vs. 15% of the control group, *p* = 0.0535), in the frequency of night symptoms (58.33% of the study group vs. 65% of the control group, *p* = 0.6511), and in the excessive use (more than two times a week) of asthma medication (*p* = 0.5463).


Medical history


The analysis also considered the patients’ medical history, showing that patients diagnosed with endometriosis and allergies more often had mothers with a positive medical history. The difference within the groups was statistically significant (*p* = 0.0002). Data are presented in the Table 3. 

## 4. Discussion

Endometriosis is an estrogen-dependent inflammatory disorder that affects approximately 5–10% of the general female population of reproductive age [14]. The pathophysiology of endometriosis is not completely understood. The most widely known etiology is Sampson’s theory, which is based on endometriosis arising as a result of the retrograde flow of menstrual discharge from the uterus through the fallopian tubes, with the spillage of endometrial cells on the ovary and other sites in the pelvis [7]. Another theory to complement retrograde menstruation is the autoimmune theory. In endometriosis, chronic local inflammatory processes are well recognized, and the presence of autoantibodies, as in some autoimmune diseases, has been observed [15]. There is also evidence that immunological aspects such as immune cells, growth factors, and cytokines may be related to the pathophysiology of infertility in endometriosis, via altering the eutopic and the ectopic endometrium [16,17]. We conducted a study to correlate the frequency of endometriosis with allergies and other autoimmune diseases, as well as with eating habits. To the best of our knowledge, this was the first study that tested a general population of women using social media and that was not influenced by hospital procedures. 

Nickel allergy is a disease whose prevalence ranges from 10 to 23% among women [18,19], and is one of the most common forms of allergic contact dermatitis. Nickel is a silvery-white metal used for plating, coins, batteries, jewelry, buttons, zippers, valves, heat exchangers, and promoters. Yuk JS. et al. proved that a nickel allergy is correlated with endometriosis [18]. Their research included 4985 patients, 997 of whom had endometriosis, of which only eight patients had nickel allergy. In the control group there were 13 nickel allergy patients. The prevalence of a nickel allergy in the endometriosis group was therefore 0.8% vs. 0.3% in the control group (*p* = 0.044) [18]. In our research, 20.98% women with endometriosis had a nickel allergy (*p* = 0.23361). Our data do not confirm that a nickel allergy was correlated with endometriosis. 

In our research, we also analyzed the incidence of food allergies—105 women with endometriosis had food allergies (51.22%) whereas among the women without endometriosis, only 36 had food allergies (44.44%; *p* = 0.3018). A similar correlation was noticed in the study by Schink et al., where the prevalence of self-reported food intolerances was significantly higher in the endometriosis patients compared with the controls (*p* < 0.009) [20]. In our analysis, variables concerned with the frequency of soya product, beef, poultry, fish, and dietary products consumption, as well as alcohol use, were considered. The differences between the study and the control groups for the consumption of poultry, beef, and fish were statistically significant, (*p* = 0.0145, *p* = 0.0143, *p* = 0.0016, respectively), with a higher consumption of beef among the endometriosis patients with allergies. The frequency of soya product consumption was correlated with endometriosis diagnosis (*p* = 0.008), and there was also a correlation between the frequency of alcohol use and endometriosis occurrence, which equaled *p* = 0.0293.

Our study showed also that the consumption of milk and dairy products among patients suffering from endometriosis with allergies was lower than among the controls. This would suggest that high milk consumption could be a protective factor for the occurrence of endometriosis. In addition, the consumption of yellow cheese was the smallest in the group of allergic patients with diagnosed endometriosis, and the difference was statistically significant (*p* = 0.0002). This is also described by Nodler et al. in their study, where they showed that the intake of total dairy foods in adolescence was associated with a lower risk of endometriosis in adulthood in the prospective analysis. After adjustment for covariates, women who consumed more than four servings per day of total dairy foods as adolescents had a 32% lower risk of laparoscopically-confirmed endometriosis compared with women consuming less than one serving/day [21]. In addition, Trabert et al., 2011, found an inverse correlation between dairy intake and endometriosis [22]; the explanation for this correlation may be the association between endometriosis and vitamin D, which may be derived from the observation that vitamin D stimulates T-regulatory cells and the secretion of IL-10, reduces the concentration of proinflammatory cytokine IL-17, and dampens T-helper 1 immune function [22,23,24,25].

Our study found that the median red meat consumption was higher in women with endometriosis with allergies than in the rest of the groups. The differences between the study and the control groups for beef consumption were statistically significant (*p* = 0.0143). In a prospective cohort study by Yamamoto et al., meat intake was associated with a higher risk of endometriosis. Women consuming more than two servings of red meat per day had a 56% higher risk of laparoscopically-confirmed endometriosis compared with women consuming less than one serving per week [26]. The same connection was demonstrated in an Italian study by Paparazzini et al., where the increased risk of developing endometriosis was associated with a high consumption of beef and other red meat and ham [27]. In contrast, in the study by Ashrafi et al., it was shown that decreased endometriosis risk was connected with four to six portions of meat per week compared with those who consumed none to three portions of meat per week [28]. 

We noticed that endometriosis was correlated with the severity of allergy symptoms. Abdominal pain other than what was typical for endometriosis was more than once a week in 60.31% of patients with endometriosis (*p* = 0.00000). In addition, 75% patients with endometriosis with allergies had problems with defecation, such as constipation or diarrhea. In our research, bowel movement changes (*p* = 0.00006) and perianal lesions (*p* = 0.03481) were more often found in patients with endometriosis. Very similar results were presented in the manuscript by Schink et al., where about 77% patients from the endometriosis group suffered from gastrointestinal symptoms—constipation, diarrhea, or frequent defecation were significantly more frequent in endometriosis patients [20]. The same information on gastrointestinal problems can be found in other available publications [29,30].

Another allergic-dependent disease that is often studied in the context of endometriosis is asthma. Many epidemiological studies have indicated that in the adult population, the prevalence, morbidity, and severity of asthma are higher in women with endometriosis [31,32]. Such conclusions led to searching for the reason for this phenomenon. The relationship between asthma and endometriosis has been explored many times already, but the evidence is still inconsistent. Ferrero et al. concluded that asthma had a similar prevalence in women with and without endometriosis [33]. On the other hand, Sinaii et al., in their research, proved that asthma occurs more often in women with endometriosis [34]. Smorgick et al. drew the same conclusions [35]. There is evidence suggesting a significant role in the pathophysiology of asthma through the overproduction of T helper (Th) cytokines by Th2 cells. More than that, in endometriosis, we can observe a possible shift towards a Th2 immune response, as demonstrated by the relative increase in Th2-related cytokines [36]. This may explain the connection between asthma and endometriosis.

Peng YH. et al., in a nationwide study of 36,685 women, indicated that women of reproductive age with asthma had a higher risk of endometriosis [37]. Their research included 371 (5.06%) of 7337 patients with asthma and 926 (3.16%) of 29,348 enrollees without asthma who also had endometriosis. The cumulative incidence of endometriosis was higher in the asthma group than in the non-asthma group (log-rank test, *p* < 0.001). Our study was less dense, and we reported that 11.71% of women with endometriosis also had asthma (*p* = 0.00611). Moreover, in 41.67% of women with endometriosis, the symptoms limited the daily physical activity. In comparison to this, 15% of women without endometriosis had symptoms of asthma that limited their daily activity (*p* = 0.05355). Nocturnal symptoms were more often found in women without endometriosis, but insignificantly (*p* = 0.65111). All this data may support the fact that the development of endometriosis was affected by asthma, but it did not intensify symptoms and did not limit daily activity.

In our study, the correlation between other autoimmune inflammatory diseases was not studied. However, according to Zervou et al., substantial abnormalities in the immune systems of women with endometriosis have been demonstrated, which suggest a dependance between endometriosis and the risk of incident rheumatoid arthritis [38]. The background of their study considered the similarities between the molecular and cellular pathways of endometriosis and rheumatoid arthritis (RA). Similar conclusions were proposed by Chen et al. [39] In their study, patients with endometriosis were associated with an increased risk of incident RA compared with the unaffected controls. This may be due to the interleukin-17 association with endometriosis and its crucial role in several inflammatory and autoimmune diseases, such as rheumatoid arthritis, which was concluded by Shi et al. [40]. Nevertheless, this problem should be investigated in further studies. 

Limitations

This study has several limitations. The first concerns the diagnosis of endometriosis in the study group. In our research, only 49.6% patients had diagnosis based on laparoscopy or laparotomy, which, according to Hsu et al., are the gold standards in endometriosis diagnosis [41]. The rest of the females had transvaginal ultrasonography, contrast-enhanced ultrasound, or magnetic resonance imaging of the lower pelvis (5.2%) performed, or were diagnosed during advanced hysteroscopy (<1%) or cesarean section (<1%). The second limitation is that the questionnaire was distributed online, which may have caused bias due to the lack of control of patients. However, the method of study group collection is also a strong benefit of this study, as online distribution of the survey allowed for more study and control groups.

## 5. Conclusions

We concluded that endometriosis did not affect autoimmune diseases and their courses. On the other hand, patients with diagnosed endometriosis presented allergy symptoms much earlier than the healthy patients. Dietary conditions such as soya products, red meat, and alcohol had an influence on the occurrence of endometriosis. 

## Figures and Tables

**Figure 1 medicina-58-01038-f001:**
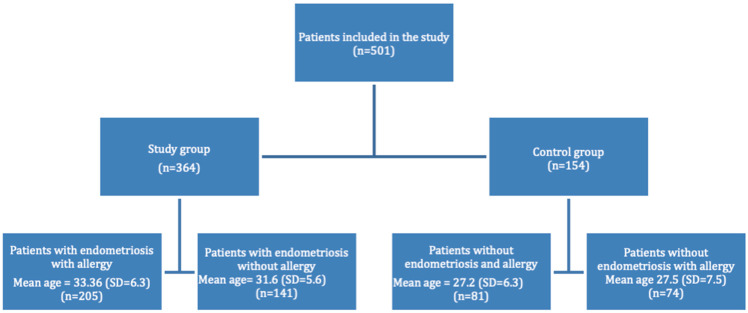
Distribution of patients between the groups in the study (SD—standard deviation, n— number).

**Table 1 medicina-58-01038-t001:** Characteristics of the groups regarding their parity status.

	Study Group		Control Group		Study Group vs. Control Group (*p* =)
Group	Endometriosis and Allergy	Endometriosis without Allergy	Healthy with Allergy	Healthy without Allergy	
Mean no. of pregnancies	0.80 (SD = 1.0)	0.80 (SD = 1.0)	0.48 (SD = 0.95)	0.55 (SD = 0.91)	0.0067
Mean no. of miscarriages	0.23 (SD = 0.59)	0.206 (SD = 0.56)	0.0864 (SD = 0.39)	0.0812 (SD = 0.32)	0.0253
Mean no. of CC	0.34 (SD = 0.63)	0.3243 (SD = 0.65)	0.2037 (SD = 0.49)	0.2642 (SD = 0.56)	0.1252
Mean no. of premature births	0.13 (SD = 0.42)	0.1081 (SD = 0.34)	0.0926 (SD = 0.29)	0.0741 (SD = 0.27)	0.6968
Mean no. of late births	0.13 (SD = 0.37)	0.1091 (SD = 0.31)	0.1481 (SD = 0.36)	0.1132 (SD = 0.48)	0.6328

**Table 2 medicina-58-01038-t002:** Gastrointestinal tract problems in the groups.

Type of Problem	Patients with Endometriosis [%]	Healthy Patients [%]	*p* Value
Abdominal pain (<6 months ago)	79.47% (*n* = 151)	41.33% (*n* = 31)	***p* = 0.000**
Abdominal pain (lasting > 1 day/week)	60.31% (*n* = 117)	26.39% (*n* = 19)	***p* = 0.0000**
Abdominal pain resolving after defecation	42.54% (*n* = 77)	25.76% (*n* = 17)	***p* = 0.0162**
Abdominal pain with a change in bowel movements	49.14% (*n* = 86)	36.49% (*n* = 27)	*p* = 0.06676
Perianal lesions	21.95% (*n* = 45)	11.11% (*n* = 9)	***p* = 0.03481**
Bowel movement changes	75.12% (*n* = 154)	50.62% (*n*= 41)	***p* = 0.00006**
Mouth lesions	27.32% (*n* = 56)	18.52% (*n* = 15)	*p* = 0.1207
Body mass loss	11.71% (*n* = 24)	9.88% (*n* = 8)	*p* = 0.658

**Table 3 medicina-58-01038-t003:** Characteristics of the medical history.

	Study Group		Control Group	
	Endometriosis and Allergy Diagnosed Patients	Only Endometriosis without Allergy Patients	Allergic Patients without Endometriosis	Non-Allergic Patients without Endometriosis
Healthy mother	82.09% (*n* = 110)	94.90% (*n* = 93)	92.59% (*n* = 50)	100% (*n* = 60)
Mother diagnosed with endometriosis	**17.91% (*n* = 24)**	5.10% (*n* = 5)	7.41% (*n* = 4)	0% (*n* = 0)

## Data Availability

From authors upon request.

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
