# Peer review of "Correlation between Endometriosis and Selected Allergic and Autoimmune Diseases and Eating Habits"

_medicina, 2022, doi:10.3390/medicina58081038_

Round 1

Author Response

Thank you for your time and significant comments. We have responded to all remarks and suggestions and we are counting on a positive reception.

1. Your remark regarding the lack of a type of diagnosis is perfectly correct. We have filled this gap in the "results'' section. We also added it the limitation section. In our research only 49.6% patients had diagnosis based on laparoscopy or laparotomy which are the gold standards in endometriosis diagnosis.

The rest of the females had transvaginal ultrasonography, contrast-enhanced ultrasound, magnetic resonance imaging of lower pelvis (5.2%) performed or were diagnosed during advanced hysteroscopy (<1%) or cesarean section (<1%). We are aware of the limitations of this data. The discrepancy in the diagnostic data may result from the different advancement of the centers. Our study included patients from cities with more than 500,000 people and cities <50,000. Nevertheless, the work was based on the questionnaire form, which as we know, has pros and cons. It allowed us to collect as much data as possible, but also blocked us from verifying the answers of the respondents.

Perhaps in the future we will manage to create a job that will be based solely on patients diagnosed in our center.

2. Our control group relied primarily on the presence of endometriosis. The survey took into account both the obstetric history and pain, which, as we know, may be the result of other diseases.

3. Survey study, despite its limitations, creates the largest research group. It was our main criterion. By this method, we were able to ask all patients exactly the same important questions that were necessary during this study. The group of patients examined only in our center would be a significant limitation and probably would not allow us to draw the conclusions presented by us.

4. We add a paragraph at the end of the study regarding the correlation of endometriosis with autoimmune diseases.

Reviewer 2 Report

Dear Authors,

 As per the abstract and the body of study, the objective of the research in abstract is to determine the dependence of endometriosis and allergies or autoimmune diseases in patients in reproductive age group/ to examine relationships between endometriosis and allergies, autoimmune disorders and eating habits in the general population of women using internet and social media.

 I am sorry, but there is a loss of continuity of thought process in the abstract and manuscript. The way the manuscript is handled is not systematic. There is a lack of background in the introduction. More references for the association of autoimmune disease/allergy with endometriosis should be presented in the introduction and some more information should be given about the methods. The Only exclusion criteria were an incomplete form.

There are a few quotes from the methods. 

"The survey was aimed at women diagnosed with endometriosis as well as women who do not suffer from endometriosis, but have allergic symptoms or are completely healthy"

"Then each patient was asked about the date of the diagnosis of endometriosis and the way of diagnosis. Questionnaires with empty answers were excluded from the further analysis to avoid bias resulting from misinterpretation.

I sincerely feel that the introduction, methods analysis, and discussion should be more systamatic.

Author Response

Thank you for your time and important comments.

We tried to respond to all comments in the manuscript.

We added exclusive criteria and changed the introduction according to your suggestions. Moreover, we added a paragraph describing the relationship between endometriosis and autoimmune diseases and a section with limitations of the study, at the end of discussion.

Round 2

Reviewer 1 Report

Upon reading very carefully the revised manuscript as well as the point by point responses to my initial comments, I recommend this article for publication.

Reviewer 2 Report

Accept with these changes